# The gut microbiome in end-stage lung disease and lung transplantation

Shuyan Zhang,[1,2] J. Casper Swarte,[1,3] Ranko Gacesa,[1,2] Tim J. Knobbe,[3] Daan Kremer,[3] Bernadien H. Jansen,[1] Martin H. de Borst,[3] TransplantLines Investigators, Hermie J. M. Harmsen,[4] Michiel E. Erasmus,[5] Erik A. M. Verschuuren,[6] Stephan J. L. Bakker,[3] C. Tji Gan,[6] Rinse K. Weersma,[1] Johannes R. Björk[1,2]

**ABSTRACT**  Gut dysbiosis has been associated with impaired outcomes in liver and kidney transplant recipients, but the gut microbiome of lung transplant recipients has not been extensively explored. We assessed the gut microbiome in 64 fecal samples from end-stage lung disease patients before transplantation and 219 samples from lung transplant recipients after transplantation using metagenomic sequencing. To identify dysbiotic microbial signatures, we analyzed 243 fecal samples from age-, sex-, and BMI-matched healthy controls. By unsupervised clustering, we identified five groups of lung transplant recipients using different combinations of immunosuppressants and antibiotics and analyzed them in relation to the gut microbiome. Finally, we investigated the gut microbiome of lung transplant recipients in different chronic lung allograft dysfunction (CLAD) stages and longitudinal gut microbiome changes after transplantation. We found 108 species (58.1%) in end-stage lung disease patients and 139 species (74.7%) in lung transplant recipients that were differentially abundant compared with healthy controls, with several species exhibiting sharp longitudinal increases from before to after transplantation. Different combinations of immunosuppressants and antibiotics were associated with specific gut microbial signatures. We found that the gut microbiome of lung transplant recipients in CLAD stage 0 was more similar to healthy controls compared to those in CLAD stage 1. Finally, the gut microbial diversity of lung transplant recipients remained lower than the average gut microbial diversity of healthy controls up to more than 20 years post-transplantation. Gut dysbiosis, already present before lung transplantation was exacerbated following lung transplantation.

**IMPORTANCE**  This study provides extensive insights into the gut microbiome of end-stage lung disease patients and lung transplant recipients, which warrants further investigation before the gut microbiome can be used for microbiome-targeted interventions that could improve the outcome of lung transplantation.

**KEYWORDS**  gut microbiome, dysbiosis, end-stage lung disease, lung transplantation, immunosuppressive medication

Lung transplantation is a life-saving therapy for patients with end-stage lung disease (ESLD), and survival has improved due to advances in immunosuppressive therapy, surgical techniques, and organ preservation. Nonetheless, chronic lung allograft dysfunction (CLAD) and high mortality limit the long-term outcomes of lung transplant recipients (LTR) (1–4).

The gut microbiome has now been linked to many diseases, including gastrointestinal, metabolic, and mental diseases (5). It plays an important role in maintaining immune homeostasis (6), and perturbing it may cause "gut dysbiosis," a condition typically characterized by the growth of pathobionts at the expense of commensal

Address correspondence to Johannes R. Björk, bjork.johannes@gmail.com.

Shuyan Zhang and J. Casper Swarte contributed equally to this article. Author order was determined by coin flip.

Rinse K. Weersma and Johannes R. Björk contributed equally to this article.

The authors declare no conflict of interest.

bacteria (7). Gut dysbiosis can worsen immune dysregulation which can result in disease development (7). The gut microbiome has been shown to affect lung immunity through endotoxins, microbial metabolites, cytokines, and hormones via a bidirectional crosstalk between the gut and lungs called the "gut-lung axis" (8, 9). Patients with lung diseases such as asthma, chronic obstructive pulmonary disease (COPD), cystic fibrosis, and tuberculosis often exhibit gut dysbiosis, which likely has a multifactorial origin that includes the use of drugs such as antibiotics and prednisolone (8, 10–12). It is, however, still unknown whether LTR recover from this gut dysbiosis after transplantation.

While previous studies have revealed gut dysbiosis both before and after kidney (13, 14) and liver (13, 15) transplantation, few studies have analyzed the gut microbiome in LTR (but see, e.g., reference 16). Similar to kidney and liver transplant recipients, LTR are on a life-long course of immunosuppressive drugs to prevent allograft rejection, and the use of immunosuppressive drugs was recently found to be associated with gut dysbiosis in kidney and liver recipients (13). To prevent rejection and consequent CLAD, LTR generally receive higher dosages of immunosuppressive drugs than kidney and liver transplant recipients (17). Moreover, because LTR are at increased risk of being infected by multidrug-resistant bacteria, they also receive more antibiotics than liver and kidney transplant recipients (18). These factors predispose LTR to gut dysbiosis, highlighting the need to characterize the gut microbiome in LTR, including the extent of gut dysbiosis and its possible association with transplantation outcomes such as CLAD.

Here, we study the gut microbiome in the setting of lung transplantation using shotgun metagenomic sequencing in 283 fecal samples collected from 224 LTR and 243 fecal samples from 243 healthy matched controls, all enrolled in the TransplantLines Biobank and Cohort Study. We analyzed microbial species and pathways, antibiotic resistance genes (ARGs), and virulence factors (VFs) using metagenomic sequencing data. In the current study, we addressed multiple questions: (i) What is the extent of gut dysbiosis before and after lung transplantation? (ii) Are different medication regimens associated with the gut microbiome after transplantation? And (iii) is the extent of gut dysbiosis associated with different CLAD stages?

## MATERIALS AND METHODS

### Study design

We recruited patients with ESLDs and LTR (Table 1 and Table 2). Their fecal samples were collected by the TransplantLines Biobank and Cohort Study (trial registration number NCT03272841). Eisenga et al. (19) have described the TransplantLines study in detail with regard to randomisation, rationale of the study design, and inclusion/exclusion criteria. As the HC in this study, we included fecal samples collected prior to the surgery from age-, gender-, and BMI-matched kidney donors in the TransplantLines cohort.

### Characterization of outcomes of lung transplantation by CLAD stages

CLAD, which develops in about 50% of LTR by 5 years after lung transplantation (20), remains a major threat for the survival of these recipients. Following the current definition of CLAD from the study of Verleden et al. (21), we first used the ratios of current FEV1 (forced expiratory volume in 1 second) divided by baseline FEV1 to group the LTR into different CLAD stages, with sample sizes 113 (CLAD0), 36 (CLAD1), 11 (CLAD2), 7 (CLAD3), and 4 (CLAD4). Because of the uneven and relatively small sample sizes, we combined samples belonging to CLAD2, CLAD3, and CLAD4 into one group called "CLAD2–4."

### Clinical and laboratory characteristics

At every study visit, a fixed set of laboratory parameters was measured and recorded into the study database with the consent of patients. Blood samples were collected in the morning after 8–12 hours of overnight fasting before the study visit and were

**TABLE 1** Demographics and anthropometrics of the cohort study

| Parameter | End-stage lung disease patients | Lung transplant recipients | Healthy controls |
|---|---|---|---|
| Number of subjects, *n* | 64 | 171 | 243 |
| Age, years (median [IQR]) | 58 (55–62) | 61 (55–65) | 57 (50–65) |
| Male sex, *n* (%) | 26 (40.6) | 91 (53.2) | 115 (47.3) |
| BMI, kg/m$^2$ | 25.0 ± 3.4 | 25.8 ± 4.5 | 26.4 ± 3.5 |

**TABLE 2** Distribution of end-stage lung disease etiology in end-stage lung disease patients and lung transplant recipients[a]

| Disease | Disease etiology, *n* (%) | |
|---|---|---|
| | End-stage lung disease patients (ESLD patients) 64 | Lung transplant recipients (LTR) 171 |
| Chronic obstructive pulmonary disease | 39 (60.9) | 67 (39.2) |
| Idiopathic pulmonary fibrosis | 8 (12.5) | 25 (14.6) |
| Pulmonary arterial hypertension | 8 (12.5) | 7 (4.1) |
| Cystic fibrosis | 2 (3.1) | 13 (7.6) |
| Alpha-1 antitrypsin deficiency | 3 (4.7) | 24 (14.0) |
| Sarcoidosis | 2 (3.1) | 3 (1.8) |
| Bronchiectasis | 1 (1.6) | 3 (1.8) |
| Pulmonary vascular disease | 1 (1.6) | 4 (2.3) |
| Others | – | 25 (14.6) |

[a]The "others" category in the table includes rare etiologies such as lymphangioleiomyomatosis, eisenmengers syndrome, and obliterative bronchiolitis.

measured by in-hospital routine assays to analyze clinical markers. Demographics and data on medication use were offered by the participants and confirmed by the patients during the study visit. Anthropometry measurements included height, body weight, and fat percentage (multifrequency bioelectrical impedance device; BIA, Quadscan 4000, Bodystat, Douglas). General medical information at the time point of transplantation was extracted from electronic hospital records.

## Sample collection and generation of gut microbiome data

The combination of collecting both cross-sectional and longitudinal samples made it possible to analyze the gut microbiome of patients with end-stage lung disease and lung transplant recipients, as well as its short- and long-term alteration. Transplantation candidates are screened intensively before transplantation. Patients up to the transplantation standard were included in the TransplantLines study. Pre-transplantation patients were followed as end-stage lung disease patients. Further study visits were done at 3, 6, 12, and 24 months after transplantation. Transplant recipients who underwent the transplantation before June 2015 were included in the cross-sectional part of the study for one study visit. The patients in the cross-sectional part of the study were not followed prospectively and were included at a time point >1 year after transplantation. In this study, the mean delay of collecting cross-sectional samples is 6 years after lung transplantation; the numbers of longitudinal samples collected at 3, 6, 12, and 24 months after transplantation are 5, 29, 34, and 17, respectively. Details of gut microbiome data formation are described in Supplemental Materials and Methods. Briefly, the generation of gut microbiome data included the steps of DNA extraction, library construction, metagenomic sequencing, and metagenomic data processing.

## Statistical analysis

All analysis was performed in R version 4.1, unless stated otherwise. The richness of gut microbiota (alpha diversity) was characterized using the Shannon index (using vegan

R package), of which the statistical significance between different groups was tested using the Mann-Whitney *U*-test. Principal component analysis (PCA) was used to visualize sample differences between groups and permutational multivariate analysis of variance (PERMANOVA) was performed to test for differences in microbial community composition (beta diversity) across groups based on the Aitchison distance. We applied centred log-ratio (clr) transformation to relative abundances of microbial species and metabolic pathways and then performed differential analysis using linear regression models (using the lm function in R) adjusting for age, sex, and BMI with false discovery rate (FDR) correction of 0.05 . We also modeled the presence/absence of antibiotic resistance genes and virulence factors using logistic regression adjusting for age, sex, and BMI (using the glm function in R) with the same FDR-correction as applied to the linear models. For the longitudinal analysis, we used linear mixed models adjusting for age, sex, and BMI and including participant ID as a random effect. This was done using the lmer function from the lmerTest R package. We used the R package emmeans to compute and compare the estimated marginal means for each end-stage disease. Samples with missing data were excluded. Details of all the methods used can be found in Supplemental Materials and Methods.

## Identification of medication regimens by unsupervised clustering

According to standardized protocol, there are seven commonly used immunosuppressive drugs: tacrolimus, ciclosporin, mycophenolic acid, azathioprine, everolimus, sirolimus, and prednisolone. Six antibiotic groups are frequently used: fluoroquinolone, penicillin, aminoglycoside, macrolide, imidazole, and sulfonamide trimethoprim. Usage of these medications in LTR is listed in Table S1, and 1/0 was used to indicate whether the patient used/did not use the medication at the time point of the study visit. As described above, data about the use of immunosuppressive drugs and antibiotics are binary, so we first computed the Jaccard index using the vegdist function from the vegan R package. We then performed a hierarchical cluster analysis with the ward.D agglomeration method using the hclust function from the stats R package. We then plotted a dendrogram (Fig. S1) showing the hierarchical relationships among samples from LTR. The height at which two samples are joined together represents the dissimilarity of their medication regimens. The larger the height, the more dissimilar their medication regimens. Samples joined together at a height of zero were on the same medication regimen. We cut the dendrogram at zero, thus identifyinf five medication regimens with at least five recipients (including a total of 139 cross-sectional samples collected from LTR after transplantation). We did not analyze medication regimens with fewer than five recipients.

## RESULTS

In this study, part of the TransplantLines Biobank and Cohort Study (18), we analyzed 526 fecal samples from 224 LTR and 243 age-, sex-, and BMI-matched samples from a healthy population as controls (HC). To characterize the composition and functional potential of the gut microbiome, each sample was profiled using shotgun metagenomics. In total, 171 samples were cross-sectionally collected post-transplantation. For a smaller subset of 86 LTR, longitudinal samples were collected: 64 pre-transplantation samples and 48 post-transplantation samples. To quantify the extent of gut microbiome dysbiosis in both ESLD patients (those waiting for lung transplantation) and LTR (those who have undergone lung transplantation), we compared the gut microbiome of these two patient groups with the HC gut microbiome. More specifically, we quantified differences in alpha diversity (Shannon diversity index), beta diversity (Aitchison distance), and individual (clr-transformed) relative abundances of microbial species and metabolic pathways. We also assessed differences in antibiotic resistance genes (ARGs) and virulence factors (VFs) between ESLD patients and LTR, including HC.

## Gut microbiome dysbiosis in ESLD patients and LTR

To assess the gut dysbiosis of ESLD patients and LTR cross-sectionally, we compared the 64 samples from ESLD patients and 171 samples collected from LTR with 243 samples collected from healthy controls. Compared with HC, the gut microbiome of ESLD patients and LTR had a lower Shannon diversity (Mann-Whitney $U$-test: $P_{ESLDvsHC} = 1.6 \times 10^{-3}$, $U = 9,766$; $P_{LTRvsHC} = 4.1 \times 10^{-11}$, $U = 28,691$; Fig. 1a) and altered microbial composition (PERMANOVA: $P_{ESLDvsHC} < 0.001$; $P_{LTRvsHC} < 0.001$). While Shannon diversity was not different between ESLD patients and LTR ($P_{ESLDvsLTR} = 0.15$; $U = 6,133$), we found that microbial composition was markedly different between the two groups (PERMANOVA: $P_{ESLDvsLTR} < 0.001$; Fig. 1b). We found that the average Aitchison distance between LTR and HC was significantly higher than the average distance between ESLD patients and HC (mean ± std: 77.57 ± 5.71 versus 73.95 ± 5.86, Mann-Whitney $U$-test: $U = 3,650$, $P = 8.6 \times 10^{-5}$; Fig. S2). Finally, the gut microbial diversity of lung transplant recipients remained lower than the average gut microbial diversity of healthy controls up to more than 20 years after transplantation (Fig. S3).

To identify the species that underlie the differences in microbial composition among ESLD patients, LTR and HC, we performed a differential abundance analysis correcting for potential confounders such as age, sex, and BMI. This analysis identified 108 species (58.1%) that were differentially abundant in ESLD patients and 139 species (74.7%) that were differentially abundant in LTR compared with HC (Fig. 1c; Table S2). For example, ESLD patients were enriched with *Bifidobacterium dentium* (FDR = $5.39 \times 10^{-6}$), *Bacteroides ovatus* (FDR = $1.36 \times 10^{-3}$), and *Parabacteroides johnsonii* (FDR = $1.10 \times 10^{-3}$), while LTR had higher clr abundances of *Blautia wexlerae* (FDR = $4.32 \times 10^{-13}$), *Rothia mucilaginosa* (FDR = $2.71 \times 10^{-13}$), and *Roseburia intestinalis* (FDR = $1.19 \times 10^{-6}$). We further found 101 (54.3%) differentially abundant species that were shared between ESLD patients and LTR. Interestingly, all 101 shared species exhibited the same directional effect in both ESLD patients and LTR. Examples of species that exhibited lower clr-abundances in HC compared with ESLD patients and LTR included *Streptococcus parasanguinis*, *Clostridium saccharolyticum*, *Clostridium symbiosum*, *Ruminococcus gnavus,* and *Hungatella hathewayi* (FDR < 0.05). These species have previously been reported to be enriched in subjects with various chronic diseases (5). In an end-stage disease specific sub-analysis, we found that COPD patients were enriched with, for example, *C. saccharolyticum* and *R. gnavus* (FDR < 0.05) compared with HC (Fig. S4; Table S3). In the same analysis, we found that patients with idiopathic pulmonary fibrosis (IPF) and pulmonary arterial hypertension (PAH) were enriched with, for example, *S. parasanguinis* and *Bifidobacterium dentium* (Fig. S4; Table S3), and *H. hathewayi* and *C. symbiosum* (Fig. S4; Table S3), respectively. This sub-analysis revealed that some species remained enriched post-transplantation even when the focal end-stage disease had been cured (Fig. S4; Table S3). We found that commensal species such as *Bifidobacterium adolescentis*, *Akkermansia muciniphila,* and *Agathobaculum butyriciproducens* (FDR < 0.05) had higher clr abundances in HC compared with both ESLD patients and LTR. Interestingly, *B. adolescentis* is a key gamma-aminobutyric acid (GABA) producer in the human gut microbiome (22), and dysfunctional GABA metabolism has been associated with depression and anxiety (22), conditions that are highly prevalent among LTR (23). In the gut microbiome of ESLD patients and LTR, we also found lower clr abundances of *A. butyriciproducens*, *Faecalibacterium prausnitzii,* and *Ruminococcus torques* (FDR < 0.05), which are butyrate producing species. Butyrate is a short-chain fatty acid (SCFA) with anti-inflammatory properties that affect host physiology, including lung health (24). Finally, we found 52 (27.9%) species that were differentially abundant between ESLD patients and LTR, many of which have been previously associated with, e.g., diabetes (5), inflammatory bowel disease (6), and alcohol-related liver diseases (15).

Analyzing a subset of ESLD patients that were followed 3, 6, 12, and 24 months after transplantation, we found that species such as *Escherichia coli*, *C. symbiosum,* and *Anaerotignum lactatifermentans* exhibited increasing clr abundances after transplantation and that these post-transplantation levels were much higher than the abundance in

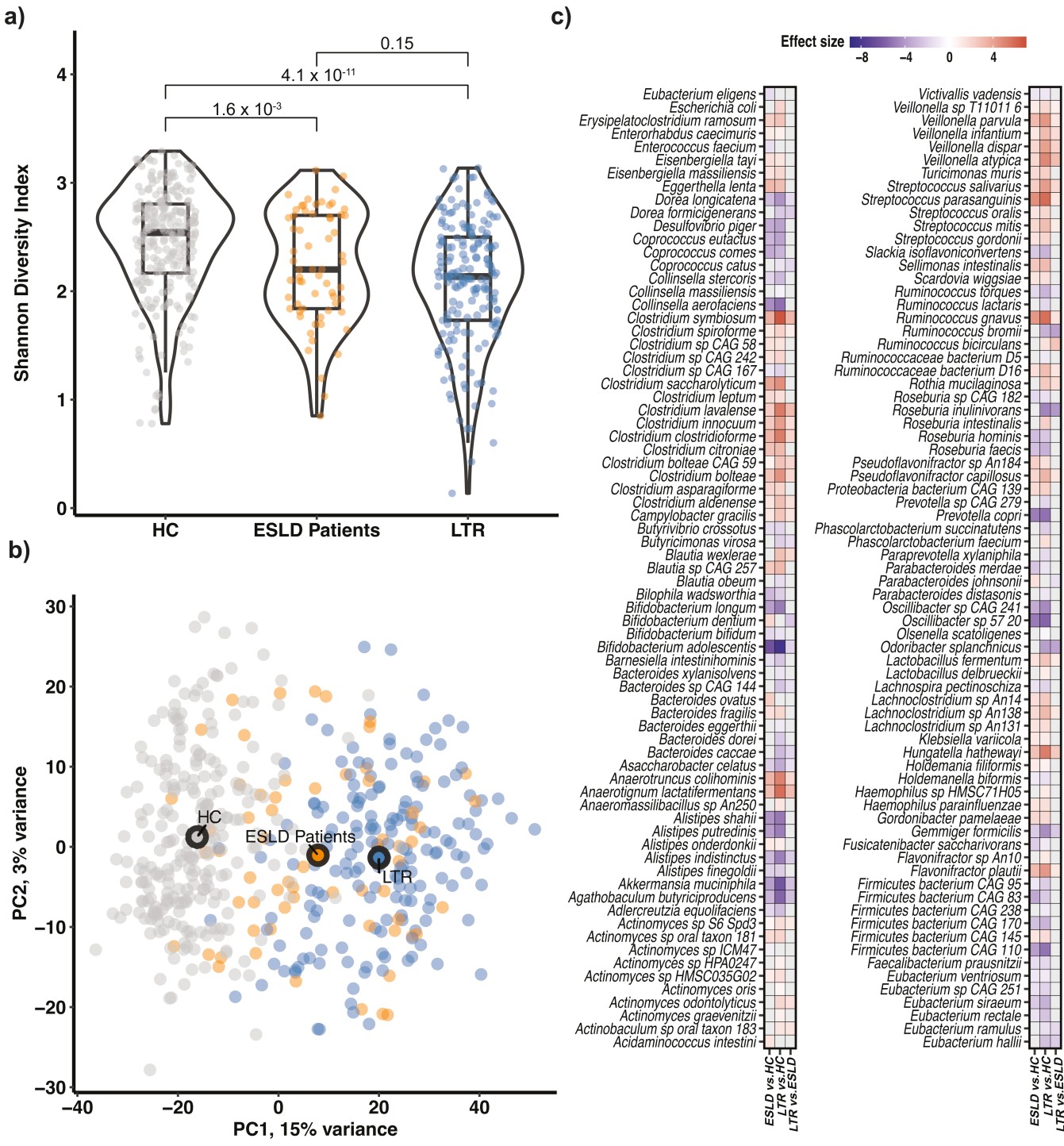

**FIG 1** The gut microbiome before and after lung transplantation is different than that of healthy controls. (a) Violin plots show the Shannon diversity index in healthy controls (HC) (gray), end-stage lung disease (ESLD) patients (orange), and lung transplant recipients (LTR) (steel blue). *P*-values are shown above each comparison. (b) PCA scatter plot with samples as dots: HC in gray, ESLD patients in orange and LTR in blue. The three larger circles represent the centroid of each group. The distance is the Aitchison distance, and samples closer to each other have more similar gut microbial community compositions. (c) Cells in the heatmap show the coefficient from a linear regression model that corresponds to the average difference (in log fold change) between either (i) ESLD patients compared with HC, (ii) LTR compared with HC, or (iii) LTR compared with ESLD patients. Cells in red and purple represent positive and negative log fold changes, respectively. Positive values indicate that the focal species has a higher abundance in the former group compared with the latter group, and vice versa for negative values. Cells in gray indicate species for which the comparison was not significant after FDR-correction.

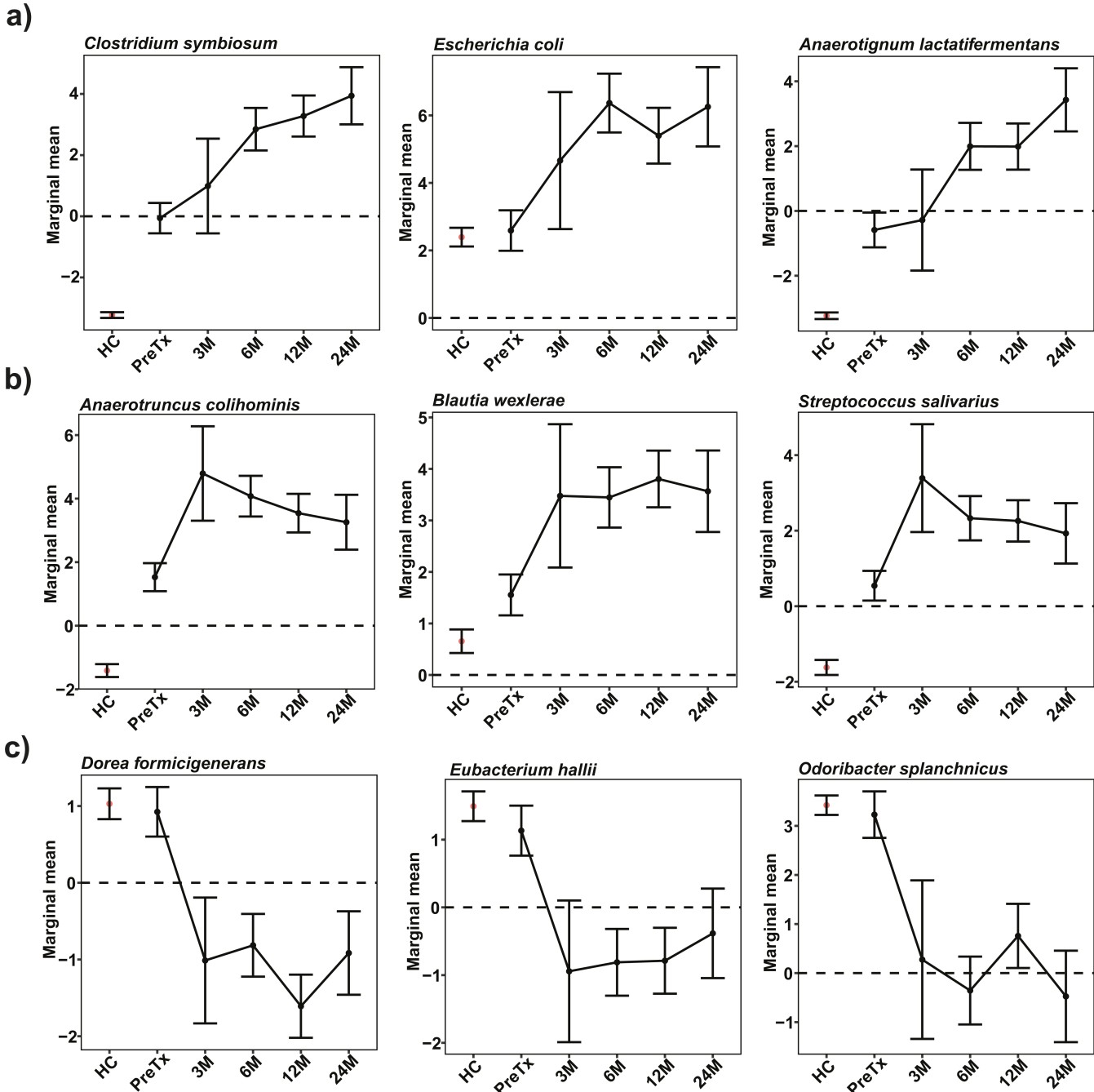

**FIG 2** Short-term dynamic changes of clr abundances of species with interesting alteration patterns. (a) Species with continuous enrichment between consecutive timepoints. (b) Species with a large expansion in three months after transplantation but either decreasing or becoming stagnant in their clr abundances as time since transplantation went by. (c) Species with a sharp decrease in three months after transplantation, of which the clr abundances in ESLD patients were similar to those in healthy controls. As a reference, each facet includes the average clr abundance (±SE) of the focal species in the healthy controls (healthy controls were not included in the linear model).

HC (Fig. 2a; Table S4). Several other species (e.g., *Anaerotruncus colihominis*, *Blautia wexlerae*, and *Streptococcus salivarius*) exhibited a large expansion in the 3 months following transplantation but then either decreased or plateaued in their clr abundances as time since transplantation increased (Fig. 2b; Table S4). Interestingly, we found several commensal species whose clr abundances were comparable between ESLD patients and

HC, but whose abundances then experienced a sharp decrease in the three months following transplantation (Fig. 2c; Table S4).

## Gut microbiome dysbiosis associated with changes in microbial metabolism

To assess whether similar changes occurred in microbial metabolism, we performed a differential abundance analysis for metabolic pathways. Here, we found 278 metabolic pathways (92%) that were differentially abundant between ESLD patients and HC (FDR < 0.05) and 279 pathways (92%) that were differentially abundant between LTR and HC (FDR < 0.05), with 269 pathways shared between ESLD and LTR compared with HC. Of the shared pathways, 77 pathways that are mostly related to amino acid metabolism, nucleotide synthesis and degradation, fatty acid metabolism, and fermentation showed lower clr abundances in ESLD patients and LTR compared with HC. For instance, three fermentation pathways showed lower clr abundances in both patient groups (pyruvate fermentation to acetone [PWY6588]: $FDR_{ESLDvsHC} = 3.02 \times 10^{-29}$, $FDR_{LTRvsHC} = 1.59 \times 10^{-52}$; glucose and fructose alteration [P124-PWY]: $FDR_{ESLDvsHC} = 1.26 \times 10^{-4}$, $FDR_{LTRvsHC} = 1.38 \times 10^{-19}$; and superpathway of acidogenic fermentation [PWY6590]: $FDR_{ESLDvsHC} = 3.28 \times 10^{-40}$, $FDR_{LTRvsHC} = 5.56 \times 10^{-70}$). PWY6590 is related to the production of acetate and butanoate, and a lower abundance of this pathway may be indicative of the dysbiosis observed in both ESLD patients and LTR. The lower abundances of some metabolic pathways in the gut microbiome of LTR can be explained by reduced abundances of related species known to possess these pathways. For example, species belonging to the phylum Firmicutes are often involved in the degradation of lactose and galactose [Lactosecat-PWY] and acidogenic fermentation [PWY6590]. Both Firmicutes members together with these two metabolic pathways had lower abundances in ESLD patients and LTR compared with HC (Table S5). Lastly, we found that the pathway responsible for acetyl coenzyme A fermentation to butanoate II [PWY5676] was enriched in the gut microbiome of both ESLD patients and LTR ($FDR_{ESLDvsHC} = 4.74 \times 10^{-15}$, $FDR_{LTR} = 5.72 \times 10^{-28}$) compared with HC.

## Antibiotic resistance genes and virulence factors more prevalent in the gut microbiome of ESLD patients and LTR

ESLD patients and LTR are susceptible to infection and are frequently treated with antibiotics, which could lead to an increased colonisation of the gut by multidrug-resistant bacteria (25). We therefore hypothesized that ARGs and VFs could be enriched in the gut microbiome of these two patient groups. To test this, we modeled the presence and absence of ARGs and VFs using logistic regression models adjusting for age, sex, and BMI. Compared with HC, we found 45 (13%) and 129 (36%) ARGs that were more prevalent in ESLD patients and LTR, respectively. Forty-two out of the 45 ARGs that were more prevalent in the gut microbiome of ESLD patients were also more prevalent in LTR (Table S6). Of these 42 genes, 17 encode for antibiotic efflux proteins, 9 are related to antibiotic target alteration, and 11 are antibiotic inactivation genes. Regarding antibiotic classes, of these 42 genes, 5 are related to the resistance of tetracycline, 5 are resistant to aminoglycoside, 4 are in relation with the resistance of macrolide, lincosamide, and streptogramin (Table S6).

VFs are genes that code for proteins associated with pathogenic mechanisms (26). Compared with HC, we found 15 (3%) and 34 (7%) VFs that were more prevalent in ESLD patients and LTR, respectively. In total, 10 VFs were more prevalent in both patient groups compared with HC (Table S7), including proteins related to bacterial adherence (VF0221, VF0404, VF0506), iron and heme uptake proteins (VF0256, VF0227), and proteins involved in bacterial invasion (VF0114, VF0221).

## Gut microbiome dysbiosis is associated with different medication regimens

Immunosuppressive drugs and antibiotics are essential to prevent allograft rejection and infection in LTR, but several studies have shown that these drugs influence the

gut microbiome (27, 28). Therefore, we next analyzed whether samples from LTR taking different combinations of immunosuppressive drugs and antibiotics differed in microbiome diversity and microbial composition. To identify the different medication regimens that were used by multiple recipients, we performed an unsupervised cluster analysis (see Materials and Methods for more detailed information). In this analysis, we focused on the seven most commonly used immunosuppressive drugs (tacrolimus, ciclosporin, mycophenolic acid, azathioprine, everolimus, sirolimus, and prednisolone) and six antibiotic groups (fluoroquinolone, penicillin, aminoglycoside, macrolide, imidazole, and sulfonamide trimethoprim) that were used by at least five LTR (Table S1). This analysis identified five clusters based on medication regimens, each with its own unique combination of immunosuppressants and antibiotics (Table 3).

LTR with each medication regimen exhibited significantly lower Shannon diversity compared with HC, with the exception of MR3 (Fig. 3a). While LTR with different medication regimens did not exhibit differences in Shannon diversity (Fig. 3a), they differed in microbiome composition. More specifically, we found that the microbial composition of MR1 users differed from MR2 and MR3 users (PERMANOVA: $P_{MR1vsMR2}=0.017$; $P_{MR1vsMR3}=0.022$), MR2 users differed from MR3 users ($P_{MR2vsMR3}=0.012$), and that MR4 users differed in microbial composition from MR5 users ($P_{MR4vsMR5}=0.044$).

We next performed a differential abundance analysis comparing all of the five medication regimens. We found the largest number of differentially abundant species between MR1 and MR2 (12 species). For example, we observed lower clr abundances of gut commensals such as *Roseburia hominis* (FDR = 0.049), *F. prausnitzii* (FDR = 5.46 × $10^{-3}$), and *B. longum* (FDR = 0.023) in MR2 users compared with MR1 users. Interestingly, the only difference between these two regimens is the use of the antibiotic group macrolide, which has previously been reported to have a strong inhibitory effect on commensal bacteria (29). We also observed lower clr abundances of *Escherichia coli* in macrolides users compared to non-users (MR1 vs MR2, FDR = 0.023; MR2 vs MR5, FDR = 0.008; MR4 vs MR5, FDR = 0.015) (30). The gut microbiome of MR1 users had lower clr abundances of several commensal species compared with MR3 users. These included, for example, *Lachnoclostridium* sp. *An 14* (FDR = 4.90 × $10^{-4}$) and *Clostridium* sp. *CAG 242* (FDR = 4.00 × $10^{-6}$; Fig. 3b; Table S8). The difference between these two regimens is the use of mycophenolic acid in MR1 but not in MR3, and the use of azathioprine in M3 but not in M1. Mycophenolic acid is hypothesized to be a stronger immunosuppressant than azathioprine, and patients treated with mycophenolic acid are more likely to suffer from gastrointestinal side effects (e.g., nausea and diarrhoea) than those treated with azathioprine (31).

**TABLE 3** Medication regimens identified by unsupervised clustering. Columns denoted MR1 to MR5 represent the five medication regimens[a]

| Drug | MR1 (N = 76) | MR2 (N = 37) | MR3 (N = 13) | MR4 (N = 8) | MR5 (N = 5) |
|---|---|---|---|---|---|
| Immunosuppressive drugs | | | | | |
| Tacrolimus | Used | Used | Used | Used | Used |
| Mycophenolic acid | Used | Used | Not used | Not used | Not used |
| Azathioprine | Not used | Not used | Used | Used | Not used |
| Prednisolone | Used | Used | Used | Used | Used |
| Antibiotics | | | | | |
| Macrolide | Not used | Used | Not used | Used | Not used |
| Sulfonamide trimethoprim | Used | Used | Used | Used | Used |

[a]The N in the parentheses shows the numbers of LTR on each medication regimen. For each regimen, green and red cells show the specific immunosuppressive drugs and antibiotics that were used and not used by LTR at the time of sample collection.

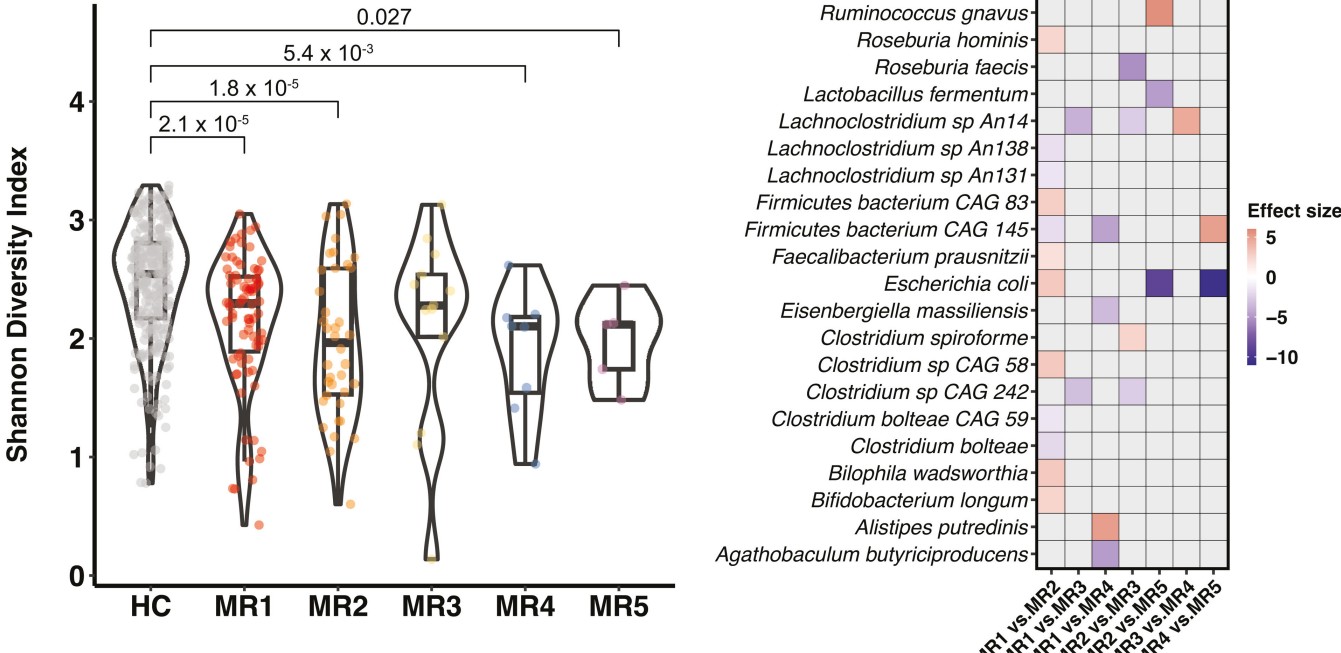

**FIG 3** Effects of different medication regimens on gut microbiome diversity and microbial composition. (a) Violin plots show the Shannon diversity index in healthy controls (HC) and lung transplant recipients (LTR) on different medication regimens (MR1–MR5, see Table 3 for details). (b) Heatmap depicts the log fold changes (i.e., effect sizes) between all pairwise combinations of different medication regimens. Cells in red represent positive log fold change and those in purple negative log fold change, which means that, compared with the latter group in each comparison, the focal species has higher (red) or lower (purple) relative abundance in the former group. Cells in gray indicate that the focal species showed no significant difference in the corresponding comparison.

## Association between gut dysbiosis and chronic lung allograft dysfunction

To test if the gut dysbiosis we observe after lung transplantation is associated with the outcome of lung transplantation as characterized by CLAD (21), we first grouped LTR into different CLAD stages ($N_{CLAD0} = 113$, $N_{CLAD1} = 36$, $N_{CLAD2} = 11$, $N_{CLAD3} = 117$, $N_{CLAD4} = 4$). Because of the uneven and relatively small sample sizes, we combined samples belonging to CLAD2, CLAD3, and CLAD4 into one group. We then quantified differences in the gut microbiome in LTR with different CLAD stages. We found no significant difference in microbial diversity among LTR with different CLAD stages (Mann-Whitney $U$, $P_{CLAD0vsCLAD1} = 0.34$, $U = 1,820$; $P_{CLAD0vsCLAD2-4} = 0.25$, $U = 1,436$; $P_{CLAD1vsCLAD2-4} = 0.091$, $U = 502$; Fig. S5a). However, LTR with CLAD0 exhibited different microbial composition compared to LTR with CLAD1 (PERMANOVA: $P_{CLAD0vsCLAD1}=0.030$; Fig. S5b). This result was corroborated by an additional analysis showing that the Aitchison distance between LTR with CLAD0 and HC was smaller than the distance between LTR with CLAD1 and HC (mean ± std: 76.77 ± 5.31 vs 79.13 ± 6.04; Mann-Whitney $U$-test: $U = 1,583$; $P = 0.046$; Fig. 4). However, no significant difference was found in the average Aitchison distance between LTR in CLAD2-4 and HC or between any of the other CLAD stages (Mann-Whitney $U$, $P_{CLAD0vsCLAD2-4} = 0.85$, $U = 1210$; $P_{CLAD1vsCLAD2-4} = 0.30$, $U = 461$; Fig. 4). Finally, we found no differentially abundant species between LTR with different CLAD stages (FDR < 0.05; Table S9).

## DISCUSSION

We studied the gut microbiome of patients with ESLD before and after transplantation and found that the gut microbiome of both patient groups exhibits gut dysbiosis when compared with HC. Overall, this dysbiosis was characterized by reduced diversity, altered compositions of microbial species, loss of important metabolic pathways, and higher prevalence of antibiotic resistance genes and VFs. Interestingly, gut dysbiosis was already present in patients with ESLD but was exacerbated after transplantation.

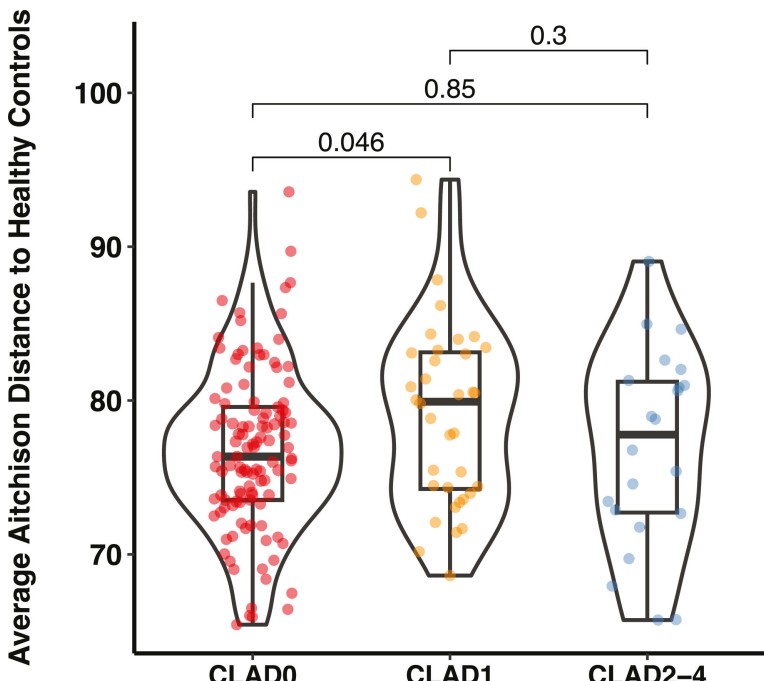

**FIG 4** Association between gut dysbiosis and chronic lung allograft dysfunction. Violin plot shows the average Aitchison distances of lung transplant recipients in different chronic lung allograft dysfunction (CLAD) stages to healthy controls.

Patients with different ESLDs all exhibited gut dysbiosis, but with different gut microbial signatures, which may be due to different antibiotic use. COPD patients showed the largest number of differentially abundant species compared with HC. This may be because COPD patients are more likely to receive antibiotics than patients with IPF or PAH. It was previously reported that the gut microbiome influences COPD progression (32) and that multiple *Streptococcus* species and members of the Lachnospiraceae family were associated with reduced lung function (33). In agreement with this, we found higher abundances of, for example, *Lachnoclostridium* sp. An138, *Streptococcus salivarius* and *S. parasanguinis* in COPD patients compared to HC. *Pseudoflavonifractor* sp. An184, which belongs to family Ruminococcaceae, had a higher abundance in patients with COPD. The abundance of this species was previously associated with insulin sensitivity and energy metabolism (34). Thus, it is possible that it plays a role, together with steroids, in the onset of post-transplantation diabetes mellitus, which is a common complication among LTR (35). We also found lower abundances of *Alistipes* spp. from the phylum Bacteroidetes, which has been associated with reduced lung function in COPD patients (36). Interestingly, most microbial signatures that we found among end-stage COPD patients were still present in LTR treated for COPD. This indicates that, while the disease is cured, the dysbiosis exhibited by the gut microbiome remains and even appears to be exacerbated after transplantation compared with before.

To date, no study has characterized the gut microbiome of IPF patients. In our study, among the different end-stage diseases, IPF patients had the smallest number of differentially abundant species compared with HC. Compared with HC, significantly increased abundances of *Streptococcus* and *Veillonella* were found in the gut microbiome of patients with each ESLD in this study (COPD, IPF, and PAH). Previous studies also reported that *Streptococcus* and *Veillonella* showed higher abundance in the gut microbiome of patients with COPD (33), PAH (37), and COVID-19 (38) compared with HC, and these species were also enriched in the lung microbiome of patients with IPF (39), COPD (40), and lung cancer (41).

Gut dysbiosis patterns of LTR in this study were similar to those previously reported for liver and kidney transplant recipients (13). First, the abundances of SCFA-producing gut commensals such as *A. muciniphila*, *Eubacterium rectale,* and *B. adolescentis* were reduced, and correspondingly, the metabolic pathways producing SCFAs were less abundant in the gut microbiome of the lung, liver, and kidney transplant recipients compared to HC (13). In addition, pathogens from genera *Clostridium* and *Streptococcus* were enriched, and these are common microbial signatures for gut dysbiosis in multiple diseases, including inflammatory bowel disease (42), alcohol-related liver diseases (15) and depression (43). The gut microbiome of LTR also showed a higher prevalence of antibiotic resistance genes and VFs, which also typically indicates gut dysbiosis (44).

Increasing evidence also suggests the existence of a gut-lung axis (8, 33), and community shifts in the gut microbiome have been correlated to the lung function decline of COPD patients (36). We hypothesized that there was a possible association between the gut microbiome and the outcome of the lung transplantation that could be characterized by the development of CLAD. We found that the average Aitchison distance between LTR in CLAD1 and HC was significantly higher than that between LTR in CLAD0 and HC, but we did not find any further significant association between the gut microbiome and CLAD stages, although this might be due to the relatively small sample sizes in our study.

Consistent with the discovery that macrolides can strongly inhibit the growth of gut commensal species (29), we observed a significant decrease of beneficial species such as *R. hominis*, *F. prausnitzii,* and *B. longum* in macrolide users compared to non-users. Korpela et al. observed that exposure to macrolides was linked to a reduction in *Bifidobacteria* of the gut microbiome of preschool children (45). The decreased abundance of *E. coli* in the gut microbiome of macrolides users compared with non-users further proves that macrolides can reduce the abundance of *E. coli*, as previously reported (30). Macrolide antibiotic, e.g., azithromycin, is often used as a treatment for bronchiolitis obliterans syndrome (BOS), one of the most severe pulmonary complications of chronic lung rejection (46). This means that LTR using macrolides are more likely to receive higher dosages of immunosuppressants, which may also explain differences between MR1 and MR2, although LTR in those two groups used the same types of immunosuppressants.

This study has several limitations. First, we did not take past exposure to treatments and medication use of lung transplant recipients, as well as the dosages of used immunosuppressants and antibiotics, into account in this study. Another limitation is the uneven sample sizes of the LTR in the different divided groups in the ESLD and medication regimen analysis. Further studies are needed in order to see whether these results are generalizable in other cohorts with larger N. Lastly, there was limited completeness in sampling in the longitudinal analysis as pre-transplant samples were not collected for all patients and not all patients provided samples at all post-transplant time points.

## Conclusions

Gut dysbiosis, already present before lung transplantation, was exacerbated following transplantation, which was associated with different use of immunosuppressants and antibiotics. Also, the gut microbiome of LTR in CLAD0 was more similar to healthy controls compared with LTR in CLAD1. This study provides extensive insights into the gut microbiome of ESLD patients and LTR. We show extensive dysbiosis in patients with ESLD, and even more extensive dysbiosis in the setting of lung transplantation, which warrants further investigation before the gut microbiome can be used for microbiome-targeted interventions that could improve the outcome of lung transplantation.

### ACKNOWLEDGMENTS

We would like to thank the Centre for Information Technology of the University of Groningen (RUG) for their support and for providing access to the Peregrine high-performance computing cluster and the Genomic Coordination Centre (UMCG and RUG)

for their support and for providing access to Calculon and Boxy high-performance computing clusters. Metagenomics library preparation and sequencing were done at MGI as part of the Million Microbiome of Humans Project (MMHP). We also thank Kate McIntyre (Scientific Editor, Department of Genetics, University Medical Centre Groningen) for English and content editing.

The TransplantLines Biobank and Cohort study received funding from Astellas BV (TransplantLines Biobank and Cohort study) and Chiesi Pharmaceuticals BV (PA-SP/PRJ-2020-9136) and was co-financed by the Dutch Ministry of Economic Affairs and Climate Policy by means of the PPP-allowance made available by the Top Sector Life Sciences & Health to stimulate public-private partnerships. R.K.W. and J.R.B are supported by the Seerave Foundation, and R.K.W is also supported by the Netherlands Organization for Scientific Research (NWO), and the EU Horizon Europe Program grant miGut-Health: personalized blueprint of intestinal health (101095470). S.Z. is supported by China Scholarship Council-UG Joint Scholarship (202106790025). The funders had no role in the study design, data collection, analysis, reporting, or the decision to submit for publication.

S.Z., J.C.S., and J.R.B. drafted the first version of the manuscript. S.Z., J.C.S., J.R.B., and R.K.W. wrote and finalized subsequent versions of the manuscript. All authors critically revised and approved the final version of the manuscript. S.Z., J.C.S., and J.R.B. performed the statistical analyses. R.G. designed and implemented metagenomic data analysis pipelines. D.K., B.H.J., M.H.D.B., TransplantLines Investigators, H.J.M.H., M.E.E., E.A.M.V., S.J.L.B. and C.T.G. collected data, assisted in study planning, and critically reviewed the manuscript.

R.G. is funded by the research grant (research collaboration between UMCG & Janssen R&D) from Janssen Pharmaceuticals and declares this grant unrelated to this paper. R.K.W. declares consulting work for Takeda, unrestricted research grants from Takeda, Johnson & Johnson, Tramedico, and Ferring and speaker fees from MSD, Abbvie, and Galapagos. The other authors declare that they have no competing interests.

## AUTHOR AFFILIATIONS

[1]Department of Gastroenterology and Hepatology, University of Groningen, University Medical Centre Groningen, Groningen, the Netherlands
[2]Department of Genetics, University of Groningen, University Medical Center Groningen, Groningen, the Netherlands
[3]Department of Internal Medicine, Division of Nephrology, University of Groningen, University Medical Centre Groningen, Groningen, the Netherlands
[4]Department of Medical Microbiology and Infection prevention, University of Groningen, University Medical Centre Groningen, Groningen, the Netherlands
[5]Department of Cardiothoracic Surgery, University of Groningen, University Medical Centre Groningen, Groningen, the Netherlands
[6]Department of Medical Pulmonary Diseases, University of Groningen, University Medical Centre Groningen, Groningen, the Netherlands

## AUTHOR ORCIDs

Shuyan Zhang http://orcid.org/0000-0001-7659-101X
Johannes R. Björk http://orcid.org/0000-0001-9768-1946

## DATA AVAILABILITY

FASTQ files are publicly available at the Sequence Read Archive (SRA) under accession number PRJNA1047900. Due to patient confidentiality, the clinical patient metadata are not publicly available but can be made available upon request: send email to datarequest.transplantlines@umcg.nl and a response will be provided within 2 weeks. This access procedure is to ensure that the clinical data are being requested for research/scientific purposes only and, thus, complies with the informed consent signed

by TransplantLines participants which specifies that the collected data will not be used by commercial parties. A STORMS (Strengthening The Organizing and Reporting of Microbiome Studies) checklist is available at https://doi.org/10.6084/m9.figshare.25450048.v1 (47).

## ETHICS APPROVAL

An informed consent form was signed by all participants before sample collection. The institutional ethics review board in the University Medical Centre Groningen (UMCG) approved TransplantLines (METc 2014/077), which is in accordance with the UMCG Biobank Regulation and adheres to the World Medical Association Declaration of Helsinki and the Declaration of Istanbul.

## ADDITIONAL FILES

The following material is available online.

### Supplemental Material

**Supplemental material (mSystems01312-23-S0001.docx).** Supplemental Materials and Methods; Figures S1 to S6.
**Tables S1 to S10 (mSystems01312-23-S0002.xlsx).** Results of differential abundance analysis and input data of medication regimens.

### Open Peer Review

**PEER REVIEW HISTORY (review-history.pdf).** An accounting of the reviewer comments and feedback.

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
