## [Reviewer comments · mSystems]

The Gut Microbiome in End-stage Lung Disease and Lung Transplantation

Shuyan Zhang, J. Swarte, Ranko Gacesa, Tim Knobbe, Daan Kremer, Bernadien Jansen, Martin de Borst, TransplantLines Investigators, Hermie Harmsen, Michiel Erasmus, Erik Verschuuren, Stephan Bakker, C. Tji Gan, Rinse Weersma, and Johannes Björk

Corresponding Author(s): Johannes Björk, Universitair Medisch Centrum Groningen

Review Timeline:

Submission Date:	December 5, 2023
Editorial Decision:	January 25, 2024
Revision Received:	March 25, 2024
Accepted:	April 3, 2024

Editor: Anthony Fodor

Reviewer(s): The reviewers have opted to remain anonymous.

Transaction Report:

DOI: <https://doi.org/10.1128/msystems.01312-23>

Re: mSystems01312-23 (The Gut Microbiome in End-stage Lung Disease and Lung Transplantation)

Dear Dr. Johannes R. Björk:

Your paper has been reviewed by an expert external reviewer. As you can see, they found the paper to be timely and well written. I have also reviewed your manuscript and concur with the reviewer that the paper is clearly written and makes an important contribution to our understanding of the long-term impact on the microbiome of transplantation. As such, if you are able to respond to these concerns, the paper should in principle be acceptable for publication in mSystems.

Although the section on the impact of different medical regimens is a strength of the manuscript, I agree with the reviewer that the manuscript would be strengthened by more explicit details and discussion of the possible confounding of your results by different pharmacological treatments.

If you decide to respond to the reviewer requests, please return the manuscript within 60 days; if you cannot complete the modification within this time period, please contact me. If you do not wish to modify the manuscript and prefer to submit it to another journal, notify me immediately so that the manuscript may be formally withdrawn from consideration by mSystems.

Revision Guidelines

Sincerely,
Anthony Fodor
Editor
mSystems

Reviewer #1 (Comments for the Author):

The authors present the results of a prospective collection of fecal microbiota samples from patients before and after lung

transplantation and compare gut microbial composition between those with lung disease to healthy controls. The authors found expected differences in those with and without lung disease although were limited in their ability to adequately distinguish post-transplant graft dysfunction, potentially due to sample size issues common in microbiome studies.

The study is well-designed, fills an important niche in the microbiome literature, and the overall manuscript is clear with appropriate analyses carried out.

Comments:

1. Please include line numbers in subsequent reviews so that comments can be more easily directed
2. Introduction: The term "pathobiont" is more accurate rather than "pathogen" and was used in the literature cited for this description of dysbiosis.
3. In the methods: Please describe how sequencing was performed (Novaseq v Hiseq for example), what the target depth of sequencing was, and what the median number of reads after sequencing was before and after quality filtering as this can influence the sensitivity of identifying AMR and virulence genes.
4. Methods: Please include as a supplement a visualization of how many stools at which times relative to transplant so that this is clear. Also, what were the windows around stool collection (as visits and samples are not exactly timed)?
5. Methods: Details from the STORMS checklist that I could not find but that I would like to have included (even if noted in prior publication from 2018):
 - a. Geographic location(s) and institutions
 - b. Dates of collection and how long patients have been followed so far
 - c. Eligibility criteria
 - d. Use of positive and negative controls during extraction, library preparation, and sequencing
 - e. Controlling for batch effects
6. Methods: Please describe standard use of antibiotic prophylaxis for lung transplant patients peri-operatively (e.g. linezolid, piperacillin-tazobactam, and posaconazole are used for all, or antibiotic selections are tailored to prior cultures in those patients with cystic fibrosis for example), for CLAD prophylaxis (such as azithromycin), and for prophylaxis after treatment for rejection (with high dose steroids, patients receive typically anti-mold prophylaxis with a triazole, trimethoprim-sulfamethoxazole, and valganciclovir for example). Perhaps some of these details are in the previously published protocol, however I couldn't find them and they should be reiterated in the supplementary methods as well.
7. Methods: Please clarify whether AMR genes were identified on short reads or contigs (I'm assuming short reads) and what stringency criteria were used for identification.
8. Results: ESLD patients, in particular those with COPD, will also have received significant immune suppression and antibiotics. I would like to see a table describing the prevalence of concurrent antibiotic and immunosuppression use in each group (ESLD and LTR) at the time of sampling.
9. Results: Table 1 may need to be separated out into 1A and 1B as it was confusing what was being described by "ESLD" in the second half of the table. A little more description in the title would make it clear (i.e. etiologies of underlying ESLD in each group). Also, were there not any patients with cystic fibrosis?
10. Table 1: There are 72 patients in the other category, more than in any other category. Please separate these out more and in the footnotes describe who are included in the other group.
11. Table 2: Please include information about the MR categories and whether there were any patterns relative to time from transplant or from rejection as some of these differences in regimens may be related to rejection or because the patient is farther from transplant.
12. Results: Rejection is not an uncommon occurrence in lung transplant and the differences in prednisolone dose would be large (a few mg per day as prophylaxis versus 60mg twice daily as treatment for example) although these would not be differentiated in this study as it stands. Please indicate whether or which samples were obtained during episodes of rejection, especially those requiring high doses of steroids or other treatments (ATG for example). Please also indicate how many of the patients included had episodes of rejection or graft failure at some point, how many died or were lost to follow-up.
13. Page 8 at the end of the first paragraph-"previously associated with diseases" is vague and should be clarified as well as having references.
14. Page 9, please specify for which antibiotic classes or types resistance was detected in addition to the mechanism described here (i.e. beta-lactams and macrolides for example)
15. In Figure S3, the description would be made more clear if it was stated that the healthy controls were measured only at one timepoint but are described as a line in the figure for comparison
16. Page 12 first paragraph. I would argue that post-LTR steroids play the largest role in LTR-associated diabetes far more than the presence or absence of an organism in the gut. Acknowledging that it may enhance the ability of steroids in the post-LTR period to cause diabetes sounds a little more realistic.
17. In the discussion, I would like to see more acknowledgement that different disease types are more likely to receive antibiotics that may explain differences in the gut. This would be true for COPD with more antibiotics given typically than IPF for example.
18. Discussion: Azithromycin is often used after a rejection episode or as BOS/CLAD treatment in sicker patients so patients on azithromycin may be receiving higher doses of immune suppression that may explain differences seen in MR1 v MR2 and so forth. If this is the case, this may need acknowledgement here.

Reviewer #1**Comment 0:**

The authors present the results of a prospective collection of fecal microbiota samples from patients before and after lung transplantation and compare gut microbial composition between those with lung disease to healthy controls. The authors found expected differences in those with and without lung disease although were limited in their ability to adequately distinguish post-transplant graft dysfunction, potentially due to sample size issues common in microbiome studies.

The study is well-designed, fills an important niche in the microbiome literature, and the overall manuscript is clear with appropriate analyses carried out.

Response 0:

We thank the reviewer for the time and effort invested in reviewing our manuscript, as well as the positive and careful evaluation of our study.

Comment 1:

Please include line numbers in subsequent reviews so that comments can be more easily directed.

Response 1:

Line numbers have been added to the revised manuscript.

Comment 2:

Introduction: The term "pathobiont" is more accurate rather than "pathogen" and was used in the literature cited for this description of dysbiosis.

Response 2:

We have replaced "pathogen" with "pathobiont" in the updated version of the manuscript. (Line number: 82)

Comment 3:

In the methods: Please describe how sequencing was performed (Novaseq v Hiseq for example), what the target depth of sequencing was, and what the median number of reads after sequencing was before and after quality filtering as this can influence the sensitivity of identifying AMR and virulence genes.

Response 3:

Metagenomic sequencing was performed with the DNBSEQ Platform at MGI, China (<https://en.mgi-tech.com/>). In this platform, the target coverage is between 30x to 100x. After sequencing and before quality filtering, the median number of sequencing reads is 54,517,354, and after quality filtering, the median number is 51,041,118. We have added this information to the "Library construction and metagenomic sequencing" and "Metagenomic data processing" in the new version of the supplementary materials.

Comment 4:

Methods: Please include as a supplement a visualization of how many stools at which times relative to transplant so that this is clear. Also, what were the windows around the stool collection (as visits and samples are not exactly timed)?

Response 4:

We have visualized this information in Figure R1 below, which is also added to the updated supplementary materials (Figure S6). The figure shows, for each lung transplant recipient (LTR), the date of lung transplantation, the date of study visit (where fecal sample was collected), and the date of rejection (if it occurred).

Figure R1. The figure depicts for each LTR (y-axis), the time of transplantation, study visit (i.e., fecal collection), and rejection. Dots on the same row indicate the different events for the same LTR. Red, green and blue dots depict the date of rejection, study visit, and transplantation, respectively.

Comment 5:

Methods: Details from the STORMS checklist that I could not find but that I would like to have included (even if noted in prior publication from 2018):

- a. Geographic location(s) and institutions
- b. Dates of collection and how long patients have been followed so far
- c. Eligibility criteria
- d. Use of positive and negative controls during extraction, library preparation, and sequencing
- e. Controlling for batch effects

Response 5:

We have now updated the STORMS checklist including the required details (<https://doi.org/10.6084/m9.figshare.25450048.v1>). We have also updated the DOI of the updated STORMS checklist in the manuscript (Line 469).

Response 5a:

All participants in this study are from the Netherlands, and this study is performed at the University Medical Center Groningen.

Response 5b:

Sample collection dates are the same as the dates of study visit, which are now depicted in Figure R1 above (Figure S6 in the supplement). The follow-up time varies from 3 months to more than 20 years.

Response 5c:

Please find the list of exclusion criteria below:

- Severe dysfunction of vital organs, especially renal insufficiency (clearance <30 ml/min)
- Active malignancy, except cutaneous basalioma. A disease-free interval or probability of recurrence is specific to a particular malignancy, but in general a remission period compatible with the risk of de novo development of the malignancy in question is considered acceptable; lung transplantation in the context of bronchioloalveolar cell carcinoma is controversial.
- Significant cardiovascular disease (applied intervention of isolated coronary artery disease is accepted)
- Significant extrapulmonary end organ damage in systemic diseases, such as autoimmune diseases, diabetes mellitus, etc.
- Other diseases or age over 60-65 years if these, in combination with other factors, can significantly adversely affect the outcome of a lung transplant
- Active, untreatable extrapulmonary infections, but also chronic, completely treatment-resistant infections
- Smoking until less than 6 months before
- Not wanting or being able to undergo long-term and intensive treatment or not being able to correct treatment compliance
- Obesity (BMI >30) or severe cachexia
- (risk of recurrence) addictive behavior, smoking, alcohol or drugs, including methadone (up to less than 1 year earlier)
- Psychological problems, such as psychosis, dementia, suicide attempts
- Symptomatic osteoporosis
- Serious locomotor tract disorder
- Symptomatic complications of glucocorticoids
- Thoracic deformity
- Prior major thoracic surgery

- HIV infection
- Hepatitis B and C
- Infection/colonization with multidrug-resistant pathogens (e.g. Burkholderia Cepacia, Mycobacterium Abscessus)

Response 5d:

We did not use positive controls during DNA extraction, library preparation and sequencing. DNA-free water was used as a negative control during DNA extraction, library preparation and sequencing.

Response 5e:

All samples were randomized onto each 96-well plate, which we send to MGI for metagenomic sequencing.

Comment 6:

Methods: Please describe standard use of antibiotic prophylaxis for lung transplant patients peri-operatively (e.g. linezolid, pip-tazo, and posaconazole are used for all, or antibiotic selections are tailored to prior cultures in those patients with cystic fibrosis for example), for CLAD prophylaxis (such as azithromycin), and for prophylaxis after treatment for rejection (with high dose steroids, patients receive typically anti-mold prophylaxis with a triazole, tmp-smx, and valganciclovir for example). Perhaps some of these details are in the previously published protocol, however I couldn't find them and they should be reiterated in the supplementary methods as well.

Response 6:

Standard antibiotic regime peri-operatively is ceftazidim. However antibiotic selections are tailored to prior cultures obtained during full pretransplant assessment in all patients. For fungal prophylaxis, we use nebulized amphotericin-B. Cotrimoxazol was used as PCP prophylaxis. We do not administrate azitromycine as prophylaxis routinely direct post-operatively, but only when CLAD is diagnosed. After treatment with methylprednisolone for acute rejection, no additional prophylaxis is started. Patients continue cotrimoxazole, and it also depends on the time post-transplant valganclovir (3 months prophylaxis; D/R +/-, D/R +/-, 1 year prophylaxis D/R +/-). (Steroid) resistance acute rejection treatment with ATG valganciclovir prophylaxis is administrated for 4 months.

We have added this information to the supplementary methods as a section of "Antibiotic use of lung transplant recipients".

Comment 7:

Methods: Please clarify whether AMR genes were identified on short reads or contigs (I'm assuming short reads) and what stringency criteria were used for identification.

Response 7:

We identified AMR genes from short reads by using the shortBRED tool `shortbred_quantify.py` (v0.9.5), with markers generated using `shortbred_identify.py` (v0.9.5) on the CARD database. As for the stringency criteria, we used the default settings: 85% clustering identity, and minimum marker length of 8 amino acids.

We added this information to the "Metagenomic data processing" section in the supplementary methods.

Comment 8:

Results: ESLD patients, in particular those with COPD, will also have received significant immune suppression and antibiotics. I would like to see a table describing the prevalence of concurrent antibiotic and immunosuppression use in each group (ESLD and LTR) at the time of sampling.

Response 8:

We looked into the use of antibiotics and immunosuppressants for both ESLD and LTR. In table R1 below, we have added the percentage of patients with concurrent antibiotic and immunosuppression use in ESLD and LTR, respectively. This table has also been added to the supplementary tables (Table S10).

Table R1. Prevalence of concurrent antibiotic and immunosuppression use in ESLD and LTR.

Group	ESLD	LTR
Number of patients	64	171
Number of patients with concurrent antibiotic and immunosuppression use	8	166
Percentage of concurrent antibiotic and immunosuppression use	12.5%	97.1%

Comment 9:

Results: Table 1 may need to be separated out into 1A and 1B as it was confusing what was being described by "ESLD" in the second half of the table. A little more description in the title would make it clear (i.e. etiologies of underlying ESLD in each group). Also, were there not any patients with cystic fibrosis?

Response 9:

We have parsed the Table 1 into two tables. **Table 1A: Demographics and anthropometrics**, and **Table 1B: Distribution of end-stage lung disease etiology in ESLD and LTR patients**.

As seen from Table 1B, there were indeed a few patients with cystic fibrosis (2 vs 13 in the ESLD and LTR group, respectively).

Table 1A: Demographics and anthropometrics.

	End-stage lung disease patients	Lung transplant recipients	Healthy controls
Number of subjects, n	64	171	243
Age, years (median [IQR])	58 (55-62)	61 (55-65)	57 (50-65)
Male sex, n (%)	26 (40.6)	91 (53.2)	115 (47.3)
BMI, kg/m ²	25.0 ± 3.4	25.8 ± 4.5	26.4 ± 3.5

Table 1B: Distribution of end-stage lung disease etiology in ESLD and LTR patients. The "Others" category in the table includes rare etiologies such as, lymphangioleiomyomatosis, Eisenmenger syndrome, and obliterative bronchiolitis.

	End-stage lung disease patients (ESLD patients) 64	Lung transplant recipients (LTR) 171
Disease etiology, n (%)		
Chronic obstructive pulmonary disease	39 (60.9)	67 (39.2)
Idiopathic pulmonary fibrosis	8 (12.5)	25 (14.6)
Pulmonary arterial hypertension	8 (12.5)	7 (4.1)
Cystic fibrosis	2 (3.1)	13 (7.6)
Alpha-1 antitrypsin deficiency	3 (4.7)	24 (14.0)
Sarcoidosis	2 (3.1)	3 (1.8)
Bronchiectasis	1 (1.6)	3 (1.8)

Pulmonary vascular disease	1 (1.6)	4 (2.3)
Others	-	25 (14.6)

Comment 10:

Table 1: There are 72 patients in the other category, more than in any other category. Please separate these out more and in the footnotes describe who are included in the other group.

Response 10:

We have added more end-stage lung diseases to Table 1B in the manuscript. The additional end-stage lung diseases included in the “others” group for lung transplant recipients (LTR) is indicated in the table legend.

Comment 11:

Table 2: Please include information about the MR categories and whether there were any patterns relative to time from transplant or from rejection as some of these differences in regimens may be related to rejection or because the patient is farther from transplant.

Response 11:

To look into this, we calculated the time difference between the study visit (i.e., fecal sampling) and the date of transplantation for each LTR that were included in the medication regimen analysis. Depending on the time difference, LTRs were categorized into five groups using the ntile() R function for quantile-based grouping. As shown in Figure R2, there is no apparent relationship between medication regimen and time since transplantation. We have added this figure to the supplement (Figure S1b).

Figure R2. Dendrogram of unsupervised cluster analysis to identify different medication regimens. Dendrogram shows the hierarchical relationship among samples from LTR. The height at which two samples are joined together represents the dissimilarity of their medication regimens. The larger the height, the more different the medication regimens. Samples joined together at a height of zero were on the

same medication regimen. The dendrogram was cut at the height of zero, and medication regimens with no less than five recipients (different color rectangles) were selected for further analysis. By quantile-based grouping, LTR were categorized into five groups based on the time interval between sampling and lung transplantation, which was indicated by the colors of labels.

Comment 12:

Results: Rejection is not an uncommon occurrence in lung transplant and the differences in prednisolone dose would be large (a few mg per day as prophylaxis versus 60mg twice daily as treatment for example) although these would not be differentiated in this study as it stands. Please indicate whether or which samples were obtained during episodes of rejection, especially those requiring high doses of steroids or other treatments (ATG for example). Please also indicate how many of the patients included had episodes of rejection or graft failure at some point, how many died or were lost to follow-up.

Response 12:

We have now acquired rejection dates, which we now present in Figure R1 (Figure S6). Among the 171 LTR, 99 recipients did not experience a rejection. Among the 72 LTR with at least one rejection recorded, 61 LTRs experienced rejection before the fecal sample collection. Among the 171 LTR in our study, 11 LTR suffered from failure of transplant. While no patients were lost to follow-up, 6 died during this period.

Comment 13:

Page 8 at the end of the first paragraph-"previously associated with diseases" is vague and should be clarified as well as having references.

Response 13:

We have now replaced "diseases" with "e.g., diabetes [5], inflammatory bowel disease [6] and alcohol-related liver diseases [15]" on the lines 255-256.

Comment 14:

Page 9, please specify for which antibiotic classes or types resistance was detected in addition to the mechanism described here (i.e. beta-lactams and macrolides for example)

Response 14:

We have now added the most frequent antibiotic classes of those antibiotic resistance genes to the line 302-304. For more details, we refer to the column "Drug Class" in Table S6.

Comment 15:

In Figure S3, the description would be made more clear if it was stated that the healthy controls were measured only at one time point but are described as a line in the figure for comparison.

Response 15:

We have added this to the legend of Figure S3.

Comment 16:

Page 12 first paragraph. I would argue that post-LTR steroids play the largest role in LTR-associated diabetes far more than the presence or absence of an organism in the gut. Acknowledging that it may enhance the ability of steroids in the post-LTR period to cause diabetes sounds a little more realistic.

Response 16:

This is a very good point. We have added a "(...), together with steroids, (...)" to the line 389.

Comment 17:

In the discussion, I would like to see more acknowledgement that different disease types are more likely to receive antibiotics that may explain differences in the gut. This would be true for COPD with more antibiotics given typically than IPF for example.

Response 17:

This is a very good point. It is true that COPD patients are more likely to receive antibiotics than patients with IPF and PAH, and this was the case in our study. We have added this to the discussion on the lines 379-381.

Comment 18:

Discussion: Azithromycin is often used after a rejection episode or as BOS/CLAD treatment in sicker patients so patients on azithromycin may be receiving higher doses of immune suppression that may explain differences seen in MR1 v MR2 and so forth. If this is the case, this may need acknowledgement here.

Response 18:

This is another good point. It is indeed possible that dose-dependent differences exist that our binary medication regimen analysis does not pick up on. We have now added this point to the discussion (Lines 432-437) and limitation (Line 442) section of the discussion.

Re: mSystems01312-23R1 (The Gut Microbiome in End-stage Lung Disease and Lung Transplantation)

Dear Dr. Johannes R. Björk:

Your manuscript has been accepted, and I am forwarding it to the ASM production staff for publication. Your paper will first be checked to make sure all elements meet the technical requirements. ASM staff will contact you if anything needs to be revised before copyediting and production can begin. Otherwise, you will be notified when your proofs are ready to be viewed.

Cover Image Submissions: If you would like to submit a potential Cover Image, please email a file and a short legend to msystems@asmusa.org. Please note that we can only consider images that (i) the authors created or own and (ii) have not been previously published. By submitting, you agree that the image can be used under the same terms as the published article. Image File requirements: TIF/EPS, 7.5 inches wide by 8.25 inches tall (at least 2,250 pixels wide by 2,475 pixels tall), minimum 300 dpi resolution (600 dpi preferred), RGB, and no figure elements, e.g., arrows or panel labels. The legend should be a short description of the image, 1-2 sentences recommended.

Sincerely,
Anthony Fodor
Editor
mSystems